



# 1 Airborne laser scanning transects over Canada's northern
# 2 forests: lidar plots for science and application

Christopher W. Bater[1], Joanne C. White[1], Hao Chen[1], Piotr Tompalski[1], Txomin Hermosilla[1],
Jonathan Boucher[2], Michael A. Wulder [1]
[1]Canadian Forest Service (Pacific Forestry Centre), Natural Resources Canada, 506 West Burnside Road, Victoria,
British Columbia, Canada
[2] Canadian Forest Service (Laurentian Forestry Centre), Natural Resources Canada, 1055 rue du Peps, Quebec City,
Quebec, Canada
*Correspondence to*: Christopher W. Bater (christopher.bater@nrcan-rncan.gc.ca)
**Abstract** Mapping vegetation is required for monitoring the condition of forest resources. Satellite data provide
information on land cover and change; however, forest structural attributes are difficult to model without additional
measurements from ground plots or airborne laser scanning (ALS, also known as airborne light detection and
ranging or lidar) instruments. Over large and inaccessible areas, such as Canada's northern and predominantly
unmanaged forests, ground plots are expensive, difficult to install, and unlikely to form a statistically valid
probability sample. An alternative means to obtain information regarding forest structure in these situations is
samples of ALS (hereafter lidar plots). Transect-based samples of ALS data can be used to provide structural
information for the calibration and validation of spatially explicit predictive modelling for wide-area mapping of
forest attributes. Here we describe and share data from the recent acquisition and processing of ALS transects across
Canada's northern forests. To date, approximately 43,000 km of ALS transects have been acquired in 2023 and
2024, with additional coverage ongoing for 2025. Acquisition flight lines were designed to sample a range of
northern forest conditions and to correspond with a concurrent ground plot sampling campaign. Airborne laser
scanning data were processed into height-normalized point clouds and reprojected to a custom Lambert conformal
conic projection to align with existing national satellite information products. More than 15 million 900 m$^2$ lidar
plots were generated from the 2023 transect dataset with point cloud metrics (i.e., area-based statistical summaries
of the ALS point cloud) calculated for each 30 by 30 m cell. Presently, the 2023 lidar plots and their associated point
cloud metrics are stored in openly available SQLite GeoPackages, with additional annual transect collections to be
added when available. To accommodate a wide range of users and applications, both comprehensive and abridged
versions of the metric databases, with 369 metrics and 40 metrics, respectively, are shared. The framework that led
to the data shared here is portable to other areas with similar information needs. The data structure used was
designed to enable updates with additional open access databases of ALS transects as data acquisition and
processing are completed. This open-access dataset constitutes a vital resource for the scientific and operational
forestry communities, offering detailed and scalable measures that bridge the gap between ground observations and



wall-to-wall satellite-based inventories. These data will support the development of enhanced wildfire fuels maps,
forest inventories, and carbon products.
**1 Introduction**
Vegetation structure underpins a range of ecological, social, and economic forest values, including timber
harvesting, carbon sequestration, biodiversity, water quality, and wildfire fuels (Haslem et al., 2011; Keith et al.,
2009; Tews et al., 2004). Medium resolution satellite remote sensing (i.e., pixels sided 10 – 100 m) has proven
effective for the wall-to-wall mapping of land cover (Hermosilla et al., 2022; Vogelmann et al., 2001), monitoring
disturbance and recovery (Hansen et al., 2014; White et al., 2017), and more recently modelling attributes such as
species (Hermosilla et al., 2024). The characterization of vegetation structure, however, can be modeled using pixel-
based remotely sensed data (Coops et al., 2021), but not with the accuracies possible using light detection and
ranging (lidar) technologies, particularly airborne laser scanning (ALS). While not an entirely fair comparison due to
differences in data costs (to the end user), level of detail captured, and collection intensity, access to simultaneous
measurements of the vertical distribution of vegetation and underlying terrain morphology (Lefsky et al., 2002),
offers critical information on forest complexity and condition that cannot be captured through other modes of remote
sensing.
Investigations related to ALS and forest measurement have been ongoing since the 1980s (Aldred and Bonnor, 1985;
Nelson, 2013), and by the early 2000s the technology was recognized as a robust tool for estimating inventory
attributes related to vegetation structure (Næsset, 2004; Reutebuch et al., 2005; Wulder et al., 2008). Given the high
cost and limited access to airborne lidar instruments in the early years, many initial investigations adopted
probability sampling approaches to efficiently obtain representative data (Wulder et al., 2012b). In contrast, today
many Canadian jurisdictions are actively collecting wall-to-wall ALS data to support the development of enhanced
forest inventories; however, data acquisitions are typically focused on managed forests in the south, leaving remote,
northern forests underrepresented (White et al., 2025). Stinson et al. (2019) define forest management status in
Canada using ownership, protection status, and tenure as these three characteristics are "…related to forest
management interests, governance and objectives in a generalized way across all Canadian jurisdictions (p. 103)."
Definitions of managed forest are different for carbon accounting purposes wherein unmanaged forests are excluded
from reporting requirements (Ogle et al., 2018). Although they are not actively managed, northern forests are critical
to the aforementioned forest values. The federal government reports on all forests, both managed and unmanaged, as
implemented through the National Forest Inventory program and communicated via the annual State of the Forests
report (Natural Resources Canada, 2023). As Canada's mean annual temperature has increased at more than twice
the global rate (Bourdeau-Goulet and Hassanzadeh, 2021), northern forests are particularly vulnerable to increased
wildfire risk (Burton, 2023; Parisien et al., 2023), further underscoring the need to improve available information for
these forests.





Although typically flown in a wall-to-wall configuration, ALS data may be collected as linear samples to extend
structural information over remote areas where continuous coverage is impractical. Wulder et al. (2012b) described
lidar sampling as a cost-effective alternative to wall-to-wall lidar acquisition for large-area forest monitoring. The
authors demonstrated that statistically sound sampling and inference methods can enable robust characterizations of
forest structure, and that integration of lidar samples with field and satellite data can enhance scalability and
precision of estimates. For example, Andersen et al. (2011) presented a methodology for estimating forest biomass
over a large area of interior Alaska. The authors used a combination of ground plots and sampled ALS transects to
achieve reasonable precision, underscoring the cost-efficiency of integrating partial airborne lidar coverage. Also
working in Alaska, Babcock et al. (2018) demonstrated that sparse lidar transects, when fused with field plots and
Landsat tree cover in a Bayesian geostatistical framework, can yield wall-to-wall biomass maps with quantified
uncertainty. Nelson et al. (2012) used an airborne profiling lidar to estimate forest biomass in Norway and found that
the results were similar to those obtained through ground surveys. Building on this logic, Margolis et al. (2015)
employed a three-phase sampling design combining ground plots, airborne profiling lidar, and ICESat-GLAS
satellite lidar data to estimate biomass across the North American boreal forest.
Wulder et al. (2012a) proposed the concept of lidar plots, wherein lidar transect data, augmented by ground plot
information, provide sample-based characterizations of forest structure. Lidar plot locations are established within
sampled lidar transect swaths at a spatial resolution matching the typical size (area) of tall tree ground plots or the
pixel size of medium spatial resolution remotely sensed data (e.g., pixels sized 400-900 m$^2$). The ALS data are
processed to generate a suite of summary statistics or metrics that characterize the point cloud within each lidar plot
(e.g., mean height, maximum height, percentiles of height). Using an area-based approach (ABA) (Næsset, 2002;
White et al., 2013), a sample of co-located ground plot measurements are then used with the point cloud metrics to
generate predictions of certain inventory attributes of interest such as height, basal area, volume, or biomass, among
others. These lidar plots, with associated metrics and attributes, may then be linked to other remotely sensed data
(e.g., optical time series) via imputation, enabling the generation of spatially exhaustive and spatially explicit models
of forest structure ultimately resulting in maps representing large areas (Coops et al., 2021; Wulder et al., 2012a)
In a proof-of-concept study, Zald et al. (2016) demonstrated how lidar plots could be used as a surrogate for ground
plots to map a suite of point cloud height (mean, standard deviation, coefficient of variation, 95[th] percentile) and
cover metrics (percentage of first returns > 2 m, percentage of first returns > mean height), as well as select forest
inventory attributes (i.e., Lorey's tree height, basal area, gross stem volume, and total aboveground biomass) for a
~38 million ha forest region in Saskatchewan, Canada for the year 2010 (corresponding to the year of ALS
acquisition). Zald et al. (2016) availed upon 1,560 km of lidar transects and a set of 4,340 lidar plots to impute point
cloud metrics directly, with the ABA forest attributes carried as ancillary variables in the plot-matching process.
Expanding on this approach, Matasci et al. (2018a) employed >25,000 km of lidar transects and 80,687 lidar plots
with Landsat surface reflectance composites to produce boreal-wide maps (~552 million ha) of the same point cloud
metrics and forest structural attributes as Zald et al. (2016) for the year 2010. Matasci et al. (2018b) expanded this
approach in both space and time, mapping forest structure annually for the entirety of Canada's forested ecosystems



(~650 million ha) for each year from 1984 to 2016. Matasci et al. (2018b) availed upon seven different lidar
acquisitions and associated lidar plots (n = 84,482) to achieve national, annual maps of forest structure, thereby
enabling characterization of structural dynamics in both disturbed and undisturbed forests over the three decade
period considered. Matasci et al. (2018b)  also used a completely independent set of lidar plots, derived from
separate lidar acquisitions to validate the imputed attributes, both spatially and temporally. Collectively, these
studies demonstrate the utility of ALS sampling and lidar plots in generating spatially and temporally rich forest
structural information at landscape to continental scales.

### 1.1  Motivation

Canada's boreal forests and the communities therein are increasingly exposed to wildfire risks (Parisien et al., 2020),
yet many northern and remote regions lack detailed vegetation inventories essential for fire behavior modeling
(Crowley et al., 2023; Parisien et al., 2020; Stinson et al., 2019). In these areas outside of the managed forest zone,
accurate information on forest structure and fuel properties is limited, constraining the capacity to assess risk or plan
mitigation strategies (Crowley et al., 2023). Further, the ongoing development of the next generation Canadian
Forest Fire Danger Rating System (CFFDRS-2025) will incorporate new data sources and requires that a new suite
of vegetation and soil attributes be modelled (Canadian Forest Service Fire Danger Group, 2021). Addressing this
data gap requires spatially explicit maps of key forest structural attributes such as canopy bulk density and canopy
base height which may be estimated using ALS (Andersen et al., 2005; Martin-Ducup et al., 2025; Moran et al.,
2020; Riaño et al., 2004), but cannot be reliably derived from satellite imagery alone (Mutlu et al., 2008; Riaño et
al., 2003) and which are equally difficult to estimate in the field (Keane et al., 2005).
To support this need, the Government of Canada via the Canadian Forest Service launched the Northern Forest
Mapping program (NorthForM). Between 2023 and 2025, this initiative is acquiring ALS transects and coincident
ground plot data (Boucher et al., 2023), with the goal of modeling fuel-related forest structure attributes for wall-to-
wall mapping using satellite imagery (Coops et al., 2021). These methods build upon earlier work by the National
Terrestrial Ecosystem Monitoring System (NTEMS), which was developed to monitor Canada's forested ecosystems
on an annual basis using consistent, nationally available datasets (White et al., 2014; Wulder et al., 2024). The
NTEMS relies primarily on medium spatial resolution satellite data (initially solely Landsat, now augmented with
Sentinel 2) time series, integrated with ALS transects and ground plots, to generate national information products
characterizing disturbance, land cover, and forest structure (Hermosilla et al., 2016). The first national lidar transect
dataset was collected in 2010 to support NTEMS product development (Hopkinson et al., 2011; Wulder et al.,
2012a), and subsequent work has shown that combining these data sources enables spatially comprehensive
estimates of both forest structure and derived attributes (Matasci et al., 2018a, b; Zald et al., 2016)

### 1.2 Objectives

Herein, we describe the acquisition and processing of ALS transect data for Canada's northern forests, and the
subsequent generation of 30 m lidar plots and ABA point cloud metrics. These data are being shared in an open





repository to support the development of models needed for generating wall-to-wall predictions of attributes relevant
for characterizing forest structure and informing forest fuels mapping.
**2 Data and methods**
**2.1 Canada's northern forests**
Canada's unmanaged northern forests represent some of the largest natural treed ecosystems on Earth. Spanning
northern Quebec, Ontario, Manitoba, Saskatchewan, Alberta, and significant portions of the Yukon and Northwest
Territories, they are largely free of large-scale industrial land uses such as forestry. Unlike managed forests to the
south, these ecosystems are shaped primarily by natural disturbances such as wildfires and insect outbreaks,
although the anthropogenic footprint is expanding in some areas (Wells et al., 2020). Tree species are cold-tolerant,
primarily within the genera *Abies*, *Larix*, *Picea*, and *Pinus*, but also include *Populus* and *Betula.* Northern forests
and treed areas are part of a larger mosaic which includes lakes, rivers, and wetlands, treeless  alpine areas, maritime
heathlands, and occasional grasslands (Brandt, 2009).
**2.2 Airborne laser scanning data acquisitions**
Planning for the 2023-2025 lidar acquisition considered previous experience with national ALS transects
(Hopkinson et al., 2011), as well as recommendations from the national airborne lidar acquisition guidelines
(Natural Resources Canada & Public Safety Canada, 2022). Acquisition specifications are summarized in Table 1.
Because of the remoteness of the area of interest (Figure 1) and the impracticality of setting up base stations, precise
point positioning (PPP) services were employed to correct global navigation satellite system (GNSS) data. The
target window for data acquisition was between 15 June and 15 September of each year, and linear mode lidar
systems were required. The ALS data were collected by private sector vendors who were awarded contracts through
the Government of Canada's competitive procurement process (Table 2). Each vendor used their own aircraft,
sensors, and systems to collect data according to the specifications outlined in Table 1.



**Table 1. Summary of ALS acquisition specifications for the 2023-2025 acquisition program.**

| Requirement | Acquisition 2023–2025 |
|---|---|
| Aggregate nominal pulse density (ANDP) | 12 pulses/m$^2$ |
| Aggregate nominal pulse spacing (ANPS) | 0.29 m |
| Footprint diameter | 0.30 m |
| Scan angle | +/-20 degrees on either side of nadir (40 degrees total field of view) |
| Horizontal datum | NAD 83 CSRS epoch 2010 |
| Height reference | Vertical datum: CGVD 2013 Geoid model: CGG2013a |
| Map projection | Universal Transverse Mercator (UTM) |
| Pulse returns | Multiple |
| Classification | 1 – Processed but unclassified 2 – Ground 3 – Low vegetation 4 – Medium vegetation 5 – High vegetation 7 – Low points (noise) 9 – Water 18 – High noise |
| Intensity Value | Normalized 16-bit values, according to the method described in the ASPRS LAS 1.4 R15 specification. |
| Data Format | LAS 1.4 R-15, Point data record format 6, compressed in LAZ |
| Swath width | 500 m (2023) or 800 m (2024 and 2025) |



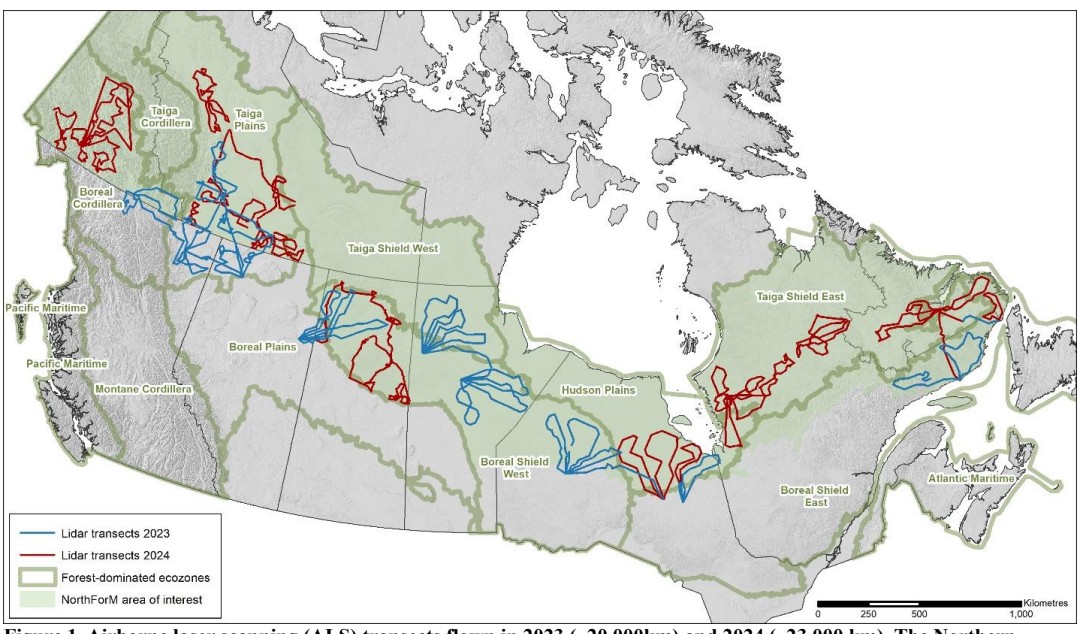

**Figure 1. Airborne laser scanning (ALS) transects flown in 2023 (~20,000km) and 2024 (~23,000 km). The Northern**
**Forest Mapping (NorthForM) acquisitions are limited to northern ecozones to improve mapping in unmanaged forests.**





**Table 2. Airborne lidar vendors for acquisition years 2023 and 2024. Each lidar plot (described in section 2.4) is linked to**
**acquisition information in a relational database.**

| Acquisition year | Vendor | Lidar sensor |
|---|---|---|
| 2023 | Aeroquest Mapcon | Riegl VQ-1560II-S |
| | Eagle Mapping | Riegl VQ-780II-S & Riegl VQ-1560II-S |
| 2024 | Aeroquest Mapcon | Riegl VQ-1560II-S |
| | Eagle Mapping | Riegl VQ-780II-S & Riegl VQ-1560II-S |
| | McElhanney | Leica TerrainMapper-2 |


Canada's National Forest Inventory (NFI) employs a systematic sampling strategy based upon 2 km x 2 km photo
plots established on a 20 x 20 km grid, with the intent to sample 1% of Canada's landmass. The 20 x 20 km sample
grid is in turn nested within a 4 x 4 km system (Gillis et al., 2005). Candidate NorthForM ground plot locations were
selected using a stratified sampling strategy employing sampling units that combined ecozone (Figure 1), and
satellite-derived percent conifer and canopy closure obtained from the Spatialized Canadian National Forest
Inventory (Guindon et al., 2024). Ground plot locations were then selected using the NFI's 4 x 4 km sampling
framework. Together, the NFI photo plot and NorthForM ground plot networks were used to guide ALS transect
design, with plot centres used as targets between which lidar data were acquired. Additional ALS transects were
established in an effort to obtain a balanced sample across northern forest-dominated ecozones where access was
possible (Figure 1).
**2.3 Data processing**
**2.3.1 Point cloud processing**
Following their delivery by the ALS vendors, subsequent processing of the point cloud data was performed using
LAStools (rapildlasso Gmbh). Footprint polygons were first created for each point cloud tile; the footprints followed
the exterior edges of ALS returns and captured large internal voids. Classified lidar point clouds were then
normalized to obtain heights above ground. Returns with scan angles exceeding 20 degrees or classified as high
noise (class 18) were dropped from the point clouds (Table 1). The point clouds were then reprojected from their
universal transverse Mercator (UTM) projections (Table 1) to a common national Lambert conformal conic
projection employed by the NTEMS program (Table 3). The normalized and reprojected point clouds were then used
to calculate point cloud metrics.





**Table 3. Projection information for National Terrestrial Ecosystem Monitoring System (NTEMS) spatial data: a custom**
**Lambert conformal conic projection with two standard parallels using the NAD83 horizontal datum. Lidar plots were**
**generated using this projection.**

| Projection information | Projected coordinate system | Lambert_Conformal_Conic_2SP |
|---|---|---|
| | Projection | Lambert conformal conic |
| | Authority | Custom |
| | Linear unit | Metre (1.0) |
| | False easting | 0 |
| | False northing | 0 |
| | Central meridian | -95.0 degrees |
| | Standard parallel 1 | 49.0 degrees |
| | Standard parallel 2 | 77.0 degrees |
| | Latitude of origin | 49.0 degrees |
| Geographic coordinate system information | Geographic coordinate system | NAD 1983 |
| | WKID | 4269 |
| | Authority | EPSG |
| | Angular unit | Degree (0.0174532925199433) |
| | Prime meridian | Greenwich (0.0) |
| | Horizontal datum | North American 1983 |
| | Spheroid | GRS 1980 |
| | Semimajor axis | 6378137.0 |
| | Semiminor axis | 6356752.314140356 |
| | Inverse flattening | 298.257222101 |


**2.3.2 Lidar plots and point cloud metrics**
Lidar plots and the databases in which they are stored were created using Python and ESRI's ArcPy package. Lidar
plots were generated as point feature classes falling within the lidar transect swaths. Using the point cloud footprints,
lidar plots were located away from the edges of swaths and large interior voids to avoid areas of missing data.  The
lidar plot centre coordinates aligned with the pixel centres of 30 m spatial resolution NTEMS raster products, which
use the NTEMS Lambert conformal conic projection (Table 3). Plots that fell within the NTEMS  land cover
product's water class (Hermosilla et al., 2022) were removed. For each lidar plot, an individual 30 m x 30 point
cloud was then clipped from which area-based metrics would be calculated in subsequent steps.
Lidar point cloud metrics were calculated for each 30 m x 30 m lidar plot using the R packages lidR (Roussel et al.,
2020; Roussel and Auty, 2023) and lidRmetrics (Tompalski, 2024). As the final products are intended to inform a
variety of applications, including forest inventory, regeneration assessment, and wildfire fuels, the metrics were
generated in four groups using: (1) all returns above 0 m, (2) first returns above 0 m, (3) all returns above 2 m, and
(4) first returns above 2 m. Two height thresholds were used so that models could be created that either consider all
vegetation from the ground surface upwards (i.e., $\geq 0$ m), or with a focus on overstory structure (> 2 m). Metrics
were calculated using only first returns as they have been shown to be more consistent than metrics based on all
returns (Bater et al., 2011); however,  metrics considering all returns provide a more comprehensive characterization
of vertical forest structure and may be preferred for applications that consider more than just the upper canopy
(Singh et al., 2016). Each group included the same set of metrics, but values varied based on the combination of
height threshold (0 m or 2 m) and return type (all returns or first returns only). In total, 369 point cloud metrics were





generated; Table 4 categorizes these metrics by type (for a full list of metrics included in the database, see
Supplement A).
**Table 4. Types of point cloud metrics calculated from non-ground returns from ALS transects. In total, 369 metrics were**
**generated. Metrics were calculated for four groups of returns using: (1) all returns above 0 m, (2) first returns above 0 m,**
**(3) all returns above 2 m, and (4) first returns above 2 m. For a full list of metrics see** Supplement **A, and for detailed**
**descriptions see Tompalski (2024).**

| Metric types | Description | Example metrics |
|---|---|---|
| Simple descriptive statistics | Basic statistical measures (e.g., mean, variance, skewness) summarizing point cloud height distribution (Bouvier et al., 2015; Lefsky et al., 2005; Nilsson, 1996). | zmean<br>zsd_above2 |
| Number of points by return number | Counts of ALS returns classified by return order. | n_return_1<br>n_return_4_above2 |
| Number and proportion of returns by echo type | The count and relative frequency of returns categorized as single, first, intermediate, or last echoes. | n_last<br>n_intermediate_above2 |
| Height percentiles | Specific quantiles (e.g., 10th, 50th, 90th percentile) of the point cloud height distribution. | zq5<br>zq50_above2_first |
| Proportion of returns above threshold height | The fraction of returns exceeding a predefined height, used to characterize canopy cover (Solberg et al., 2006). | pzabove2<br>pzabovemean_first |
| Vertical structure | Metrics describing the distribution and variation of ALS returns along the vertical axis (van Ewijk et al., 2011; Shannon, 1948). | ziqr<br>VCI_above2_first |
| Cumulative point density | The cumulative proportion of returns found in nine equal height intervals (Woods et al., 2008). | Zpcum1<br>zpcum5_above2_first |
| L-moments metrics | Statistical measures capturing the shape of the height distribution, providing robust alternatives to conventional descriptive statistics (Frazer et al., 2011). | Lcoefvar<br>L1_above2 |
| Metrics based on leaf area density | Estimates of foliage distribution and density (Hopkinson et al., 2013; Magnussen and Boudewyn, 1998) . | lad_mean<br>lad_min_above2 |
| Interval metrics | Metrics derived from predefined height intervals, summarizing point density at different canopy levels. | pz_1_2<br>pz_8_9_first |
| Rumple | A measure of canopy surface roughness or complexity based on the ratio of 3D to 2D surface area (Kane et al., 2010). | rumple<br>rumple_above2_first |
| Metrics based on kernel density estimation | Metrics derived from smoothed height distributions (McGaughey, 2024). | kde_peak3_elev<br>kde_peak2_diff_above2_first |




**2.4 Lidar plots database**
Lidar plots and associated point cloud metrics are distributed as SQLite GeoPackages[1], which are an open and non-
proprietary format. Each acquisition year (i.e., 2023, 2024, and 2025) will be stored in a separate database. Each
GeoPackage contains a point feature class storing lidar plots on the NTEMS 30 m grid, a feature class delineating
point cloud footprints, as well as a series of data tables storing point cloud metrics, province or territory, UTM zone,
ecozone, and information related to individual acquisitions (Figure 2). Given the large number of metrics in the full
database (Supplement A), for each year an abridged version of the GeoPackage is also being shared that contains a
subset of commonly used metrics for forest inventory (Supplement B).

---

[1] https://www.geopackage.org/








**Figure 2. Entity relationship diagram describing the structure of the lidar plots file geodatabase. In total, the plot metrics table includes 369 point cloud metrics for each lidar plot, with an abridged version of the database available including a subset of 40 metrics.**





**3 Results**

**3.1 ALS transects acquisitions**

A total of ~20,000 km and ~23,000 km of lidar transect data were acquired in 2023 and 2024, respectively (Figure 1). The 2023 acquisition focused on collecting data over forest-dominated ecozones that are currently lacking lidar coverage (White et al., 2025). The 2023 ALS acquisitions were significantly impacted by smoke caused by unprecedented wildfire activity in Canada (Jain et al., 2024), and as a result, 5,000 km of planned acquisitions were postponed for capture in 2024. The 2024 transects focused on acquiring data over NorthForM ground plots (Boucher et al., 2023), with ~650 plots captured. Table 5 summarizes sampling intensity within NTEMS treed land cover classes (Hermosilla et al., 2022) by ecozone (Figure 1).





**Table 5. Sampling intensity within treed land cover classes by ecozone for 2023. "Land cover pixel area (ha)" represents the area classified as a given land cover within the ecozone (Figure 1). "Land cover pixel area (%)" is the percent coverage of a given land cover type in an ecozone. "Lidar plot area (ha)" represents the area of lidar plots within the ecozone that falls within a given land cover type. "Sampling intensity (%)" is calculated as lidar plot area divided by pixel area and multiplied by 100.**

| Ecozone | Land cover class | Land cover pixel area (ha) | Land cover pixel area (%) | Lidar plot area (ha) | Sampling intensity (%) |
|---|---|---|---|---|---|
| Boreal Cordillera | Wetland-treed | 656,907 | 1.5 | 2,609 | 0.3972 |
| | Coniferous | 21,292,772 | 47.9 | 79,718 | 0.3744 |
| | Broadleaf | 1,286,953 | 2.9 | 2,915 | 0.2265 |
| | Mixedwood | 729,463 | 1.6 | 1,113 | 0.1526 |
| Boreal Plains | Wetland-treed | 5,732,402 | 8.0 | 7,930 | 0.1383 |
| | Coniferous | 17,817,472 | 25.0 | 15,142 | 0.0850 |
| | Broadleaf | 13,063,662 | 18.3 | 5,860 | 0.0449 |
| | Mixedwood | 2,104,651 | 2.9 | 2,437 | 0.1158 |
| Boreal Shield East | Wetland-treed | 1,787,152 | 1.4 | 4,888 | 0.2735 |
| | Coniferous | 42,287,435 | 34.2 | 99,850 | 0.2361 |
| | Broadleaf | 8,328,982 | 6.7 | 2,115 | 0.0254 |
| | Mixedwood | 23,206,039 | 18.8 | 23,272 | 0.1003 |
| Boreal Shield West | Wetland-treed | 3,803,299 | 4.6 | 35,432 | 0.9316 |
| | Coniferous | 24,556,792 | 30.0 | 209,945 | 0.8549 |
| | Broadleaf | 2,946,598 | 3.6 | 8,100 | 0.2749 |
| | Mixedwood | 18,467,937 | 22.5 | 90,821 | 0.4918 |
| Hudson Plains | Wetland-treed | 13,322,381 | 30.6 | 27,665 | 0.2077 |
| | Coniferous | 2,970,087 | 6.8 | 10,084 | 0.3395 |
| | Broadleaf | 112,246 | 0.3 | 396 | 0.3526 |
| | Mixedwood | 1,107,734 | 2.5 | 5,939 | 0.5362 |
| Taiga Plains | Wetland-treed | 2,291,152 | 3.7 | 30,805 | 1.3445 |
| | Coniferous | 24,969,142 | 40.3 | 163,272 | 0.6539 |
| | Broadleaf | 2,721,976 | 4.4 | 28,823 | 1.0589 |
| | Mixedwood | 886,926 | 1.4 | 5,993 | 0.6757 |
| Taiga Shield East | Wetland-treed | 210,365 | 0.3 | 1 | 0.0005 |
| | Coniferous | 28,408,741 | 36.0 | 6,259 | 0.0220 |
| | Broadleaf | 192,614 | 0.2 | 1 | 0.0005 |
| | Mixedwood | 493,404 | 0.6 | 6 | 0.0012 |
| Taiga Shield West | Wetland-treed | 361,229 | 0.6 | 237 | 0.0656 |
| | Coniferous | 17,872,110 | 29.9 | 45,534 | 0.2548 |
| | Broadleaf | 865,552 | 1.4 | 1,441 | 0.1664 |
| | Mixedwood | 741,346 | 1.2 | 853 | 0.1151 |



### 3.1.2 Quality assurance results

Overall, the ALS acquisition specifications (Table 1) were met and often exceeded. A rare exception, however, were periodic changes in footprint sizes, swath widths, and point densities in areas with complex topography. These deviations are not unexpected and occur mostly in the mountainous areas of western Canada above the tree line, and impact less than one percent of the transect data.

The ALS vendors corrected GNSS data using PPP and all reported sub-metre horizontal and vertical accuracies. Areas where transects overlap tended to have vertical differences in their digital terrain models (DTM) of several decimetres. Point cloud classifications were validated by randomly selecting 20 x 20 m areas which were then clipped to perform three-dimensional checks. Point clouds were also rasterized based on return class (Table 1) and hillshades were generated from the DTMs. Raster surfaces were then visually inspected to ensure specifications were met (e.g., water was properly classified, DTMs were representative of the bare-Earth surface). Similarly, return counts and scan angles were rasterized to ensure transects fell within the specifications for point densities and swath widths (Table 1).

### 3.2 Lidar plots databases

For the 2023 ALS transects, 15,353,866 lidar plots were generated within the lidar swaths. The full database including 369 point cloud metrics is 60.2 GB in size, and the abridged version of the database containing a subset of 40 metrics is 7.2 GB. Both versions are shared as SQLite GeoPackages.

### 3.3 Point cloud metrics

Point cloud metrics were processed in four groups using: (1) all returns above 0 m, (2) first returns above 0 m, (3) all returns above 2 m, and (4) first returns above 2 m. Figure 3 shows an example of the four processing groups from same lidar plot. The number of returns range from 19,281 (first returns > 2m) to 57,984 (all returns > 0m), while height percentiles change by varying degrees between each group. The lower height percentiles are most sensitive to changes in height threshold, with the first return P5 changing from 0.06 m (0 m threshold) to 5.71m (2 m threshold), while P95 changes from 29.91 m (0 m threshold)  to 30.55 m (2 m threshold).



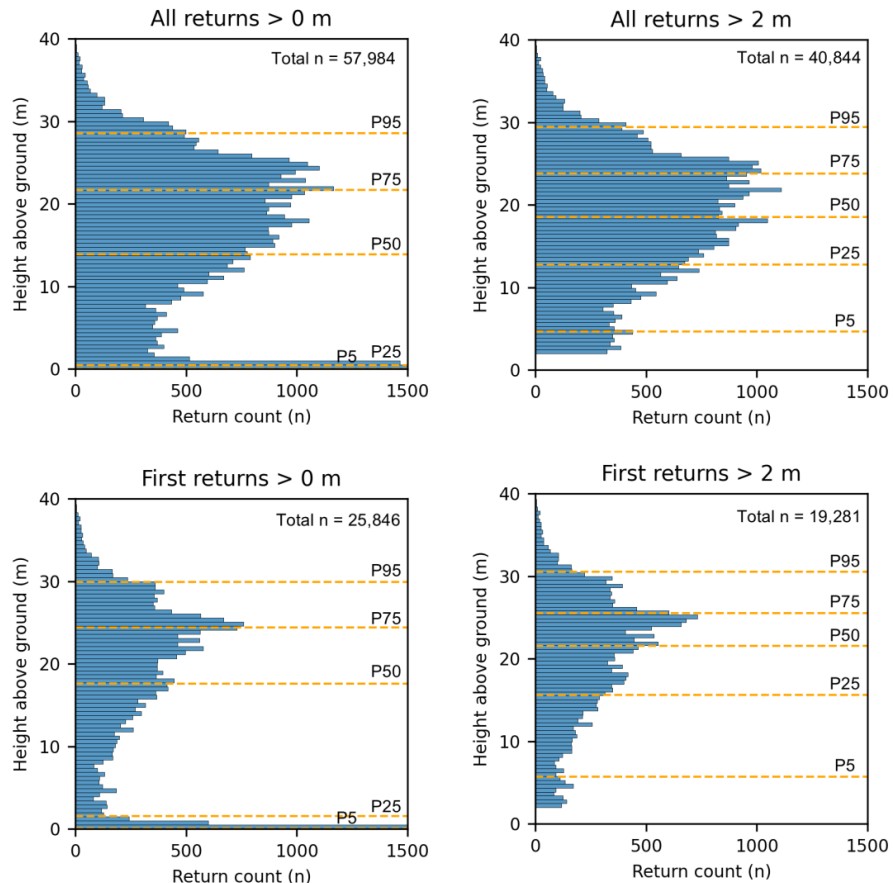

272

**Figure 3. Comparison of vertical distributions of returns from four different processing groups for the same plot: all returns above 0 m, all returns above 2 m, first returns above 0 m, and first returns above 2 m. P95 = 95th height percentile, P75 = 75th height percentile, and so on. The plot is located along the Prophet River in northern British Columbia (58º 17' 19" N, 122º 52' 30" W).**

277

Fundamentally, lidar characterizes vegetation height, vertical structure, and cover (Li et al., 2008). Figure 4 shows examples of lidar plots with point cloud metrics related to these attributes along a reach of the Liard River in Northern British Columbia. Figure 5 provides summaries of height, cover and structure by ecozone for all 2023 lidar plots.





**Figure 4. Examples of lidar plot metrics, including: canopy height based on the 95th height percentile of first returns**
**greater than 2 m; canopy cover based on the proportion of first returns greater than 2 m; and canopy complexity based**
**on the coefficient of variation of first returns heights greater than 2m. The image in the top panel extends beyond the lidar**
**swath for added landscape context. The terrain model hillshade was derived from ALS returns with scan angles in excess**
**of 20 degrees, while lidar plots are limited to returns with scan angles less than or equal to 20 degrees (Table 1). Data are**
**located along the Liard River in northern British Columbia (59º 53' 22" N, 128º 19' 3" W).**



**Figure 5: Summary of vegetation metrics by ecozone (Figure 1) for the 2023 acquisition (total n = 15,353,866 lidar plots). For the box and whisker plots, the box represents the interquartile range with the centre line showing the median, while the whiskers represent the 5th and 95th percentiles.**



### 3.3.1 Comparison of lidar plots with NTEMS satellite information products


The NTEMS project provides a number of satellite-derived products characterizing forest-dominated ecozones,
including land cover (Hermosilla et al., 2022), dominant tree species (Hermosilla et al., 2024), and recent wildfire
disturbance history (Hermosilla et al., 2016). Figure 6 provides examples of point clouds clipped to lidar plots in
three different treed land cover types. The broadleaf and coniferous plots are located in productive riparian stands,
while the wetland-treed plot is located in a nearby treed bog or fen.

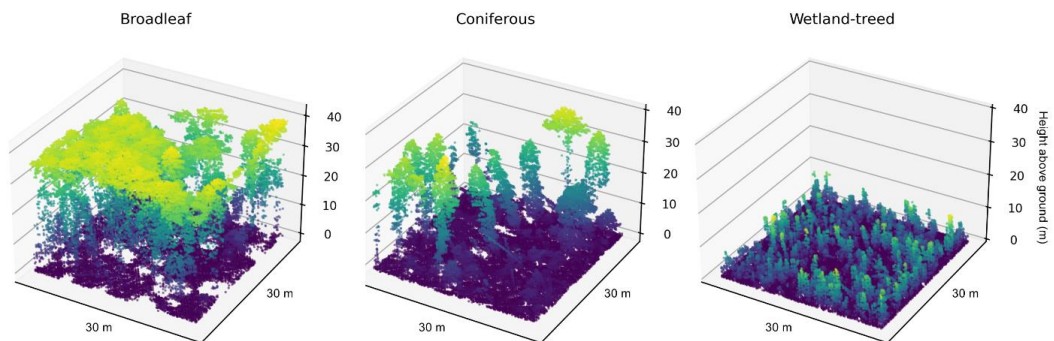


**Figure 6. Examples of point clouds within lidar plots for three different treed land cover types. The plots are located along the Prophet River in northern British Columbia (58º 17' 19" N, 122º 52' 30" W).**



Figure 7 provides distributions of 2023 lidar plots for land cover and year of recent wildfire disturbance (1985 -
2022). The dominant land cover type (Hermosilla et al., 2022) excluding water within the plots (n = 15,353,866) was
coniferous (46%), followed by wetland (17%), shrubs (11%), mixedwood (9%), wetland-treed (8%), broadleaf (4%),
exposed/barren land (3%), herbs (1%), bryoids (0.3%), rock/rubble (0.04%), and snow/ice (0.001%). Of the lidar
plots from all land cover types excluding water (n = 15,353,866), 19% were disturbed by wildfire (Hermosilla et al.,
2016) between 1985  and 2022 (Figure 7).






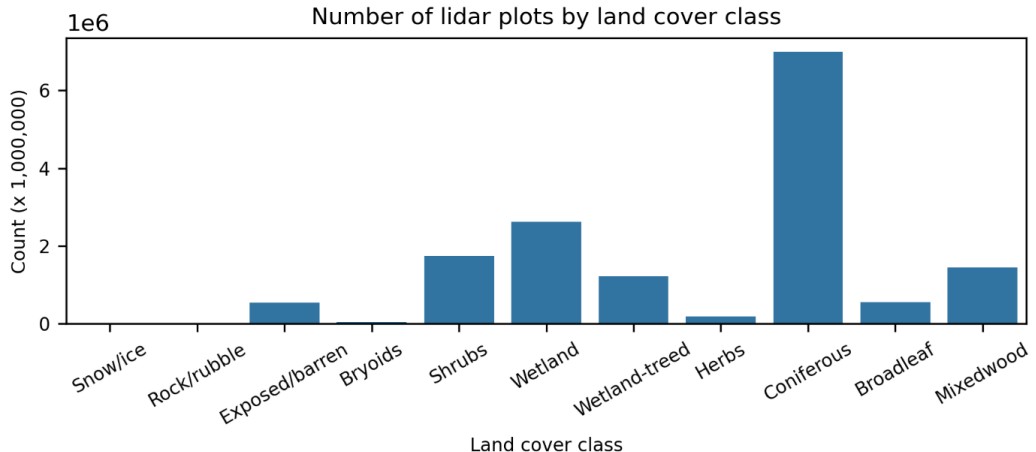

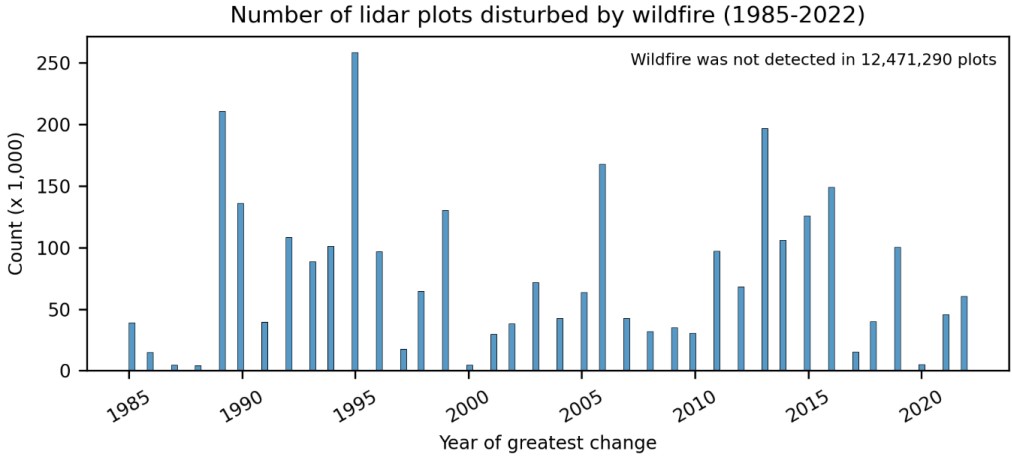


Figure 7. Comparison between lidar plots and multidecadal NTEMS satellite information products.







## 4 Discussion

**4 Discussion**
The ALS transects, lidar plots, and point cloud metrics presented here represent a comprehensive and coordinated
effort to sample forest structure in Canada's unmanaged northern forests. By collecting high-density ALS data
across ecologically diverse regions that lack structural information, this dataset fills a critical gap in the national
forest monitoring landscape. The design and implementation of the acquisitions can address both scientific and
operational needs, with particular relevance to wildfire fuel mapping, forest inventory, carbon accounting, and
ecosystem monitoring.
Open datasets allow fire researchers and other specialists unfamiliar with ALS point cloud processing to access these
data in an analysis-ready and easy-to-use format. We chose to package the data SQLite GeoPacakages, using vector
feature classes to store spatial data. The aim is that the data should be readily accessible and easy to use for those
familiar with geographic information systems or scientific programming language such as Python, R or Julia.  While
ALS derivatives are typically distributed using raster formats (e.g. Assmann et al., 2022), the layout of the transects
(Figure 1) would result in raster surfaces consisting largely of "no data" values. Should a user desire, the point
feature classes can be easily rasterized for inclusion in an analysis workflow requiring gridded surfaces. For users
interested in leveraging NTEMS datasets, the lidar plots will integrate seamlessly as all data share a common grid,
projection (Table 3), and origin coordinates.
A key advantage of this dataset lies in its flexibility. The inclusion of point cloud metrics from the four combinations
of return types and height thresholds (all returns and first returns, > 0 m and > 2 m) supports diverse modeling
approaches, including forest inventory, regeneration assessment, and canopy fuel characterization (Table 4, Figure 3,
Supplement A, Supplement B). For those focused on developing forest inventories, point cloud metrics based on
returns above 2 m, which remove the effects of shrubs and small trees, may be the most appropriate. For users
interested in forest regeneration or fuels attributes such as canopy base height, retaining lower returns may be
beneficial (Arumäe and Lang, 2018; Naesset, 2011; Stefanidou et al., 2020). The decision to use first returns or all
returns may be guided by examining performance diagnostics from predictive models (Arumäe and Lang, 2018;
Bater et al., 2011).
The value of lidar plots lies in their role as a scalable intermediary between field measurements and satellite-based
inventories, effectively increasing the sample size of required model inputs. When integrated with ground plots and
satellite data, lidar plots can enable the generation of wall-to-wall maps of forest attributes such as height, volume,
and biomass. This approach has been demonstrated nationally using earlier ALS transects (Matasci et al., 2018a, b)
and the expansion of this sampling framework substantially increases coverage across previously unsampled areas.
Despite these strengths, several aspects warrant consideration. In particular, the ALS acquisitions are restricted to
northern forests. Given the focused sampling to these northern forests, conditions present in the south will not be
captured, as exemplified by the distributions of land cover classes within lidar plots (Figure 7) differing markedly
from the national summaries reported by Hermosilla et al. (2022). These differences point to limitations of the





transects for developing national predictive models of forest structure, with a need to obtain additional samples to
represent managed forests via partnerships with provincial agencies or other accessible sources of ALS data (White
et al., 2025). Sampled transects also inhabit an unfamiliar form and scale for most users of ALS data. Within the
transects can be found detailed characterizations of both vegetation structure and terrain morphology (Figure 4,
Figure 6). The data can also be analyzed at regional scales (Figure 5) to contribute to population estimates of
attributes such as volume or biomass (Andersen et al., 2011; Margolis et al., 2015). However, transect data alone are
not spatially exhaustive, precluding independent wall-to-wall mapping and requiring the incorporation of satellite or
other ancillary data and modelling methods such as imputation (Coops et al., 2021).
One of the objectives of the NorthForM program is the collection of coincident ALS and ground plat data. As the
program progresses, GNSS locations from ground plots will be used to clip ALS point clouds to their extents. The
same suite of 369 metrics described above (Table 4, Supplement A) will then be generated for the ground plots. In
combination, the forest inventory measurements made in situ within ground plots, ground plot point cloud metrics,
and the lidar plot point cloud metrics will be powerful datasets for the spatially explicit predictive modelling of
forest structure (Matasci et al., 2018a, b; Zald et al., 2016).
Herein we focus largely on point cloud metrics derived from ALS data acquired in 2023; however, data collected in
2024 and 2025 will be made available and will follow the same processing stream and use the same basic database
schema described above. The addition of terrain metrics (e.g. height, slope, solar radiation) is underway and will be
included as an additional table in future releases.
**5 Data availability**
The 2023 lidar plots and point cloud metrics described here are available at
https://doi.org/10.5281/zenodo.16782860 on Zenodo (Bater et al., 2025).
The 2023 data and collections from subsequent acquisition years collected under the same monitoring framework
will be released as independent datasets and will share a common structure and repository. They will be made
available through Canada's National Forest Information System at: https://opendata.nfis.org/mapserver/nfis-
change_eng.html
**6 Conclusion**
The lidar plots and point cloud metrics described here form part of an open-data initiative to enhance structural
information on Canada's northern forests. By sampling remote and underrepresented forest-dominated ecozones,
this dataset supports key applications in forest inventory, wildfire risk assessment, and ecosystem monitoring. These
data offer a scalable foundation for integrating field and satellite observations to inform national mapping and
monitoring efforts, helping address long-standing data gaps in Canada's forest information landscape. In
combination with similar lidar plots representing conditions in southern Canada, these data form a key input towards



updating and improving the structural data layers (e.g., biomass, canopy height and cover) delivered via the National
Terrestrial Ecosystem Monitoring System. The inclusion of a wide range of metrics provides flexibility for diverse
predictive modeling needs, while the database structure ensures usability by researchers and practitioners who may
not be well-versed in remote sensing.
**Author contribution**
Conceptualization by MW, JW, TH, and CB. Data curation by CB. Formal analysis by CB. Methodology by JW, CB,
HC, and PT. Software by CB, HC, and PT. Supervision by MW, JW, and TH. Writing by CB, MW, JW, TH, PT, and
JB.
**Competing interests**
The contact author has declared that neither they nor their co-authors have any competing interests.
**Acknowledgements**
The ALS data were acquired with funding from the Canadian Forest Service's Northern Forest Mapping
(NorthForM) program, which aims to enhance mapping of Canada's northern forests, identify wildfire hazards, and
support community wildfire resilience and mitigation measures.

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
