# Peer review of "forests: lidar plots for science and application"

_Earth System Science Data, 2025_

## Author Comment (AC3)

The authors present a dataset collected from ALS transects with lidar-derived metrics covering Canada's northern forests. A series of 30 by 30 m lidar metrics was derived to accommodate a wide range of forestry and ecological applications. The manuscript is complete and well-written. However, there are critical aspects missing in order to make the provided datasets useful for a broad community.

**Thank you for your comments – they are very much appreciated! As the window for community discussion closes on December 1$^{st}$ our responses are brief, but they will be expanded upon in the final manuscript where appropriate.**

(1) There is no use case demonstrated, illustrating how the provided datasets can be used for "science and application" as the authors stated in the title, such as wildfire fuel mapping, ecosystem monitoring, etc. I think this is an essential aspect of demonstrating how useful this dataset is.

**As this is a data paper and not focused on specific applications, we present examples of how similar data have been used in the past and may be in the future. Please see the following text from the manuscript:**

[revised manuscript text omitted]

(2) Similarly, 369 lidar metrics have been calculated and provided; however, it remains unclear to users how to select the relevant/useful/robust metrics for different applications. Concrete examples and suggestions/usage notes should be provided on how to potentially best apply the provided lidar metrics.

**The 369 metrics are not curated for any specific application; rather our objective was to provide as many potential variables as possible for any user, thus negating the need for additional processing of the raw ALS point cloud data. We leave it to the user to determine which metrics best suit their application. That said we do provide two databases: one with the full suite of 369 metrics, and a second abridged version with 40 metrics commonly used in forestry applications. We leave it to users to determine which database best meets their needs, expecting the 40 common metrics are sufficient for most forest applications.**

**In the manuscript text, we write: *"Given the large number of metrics in the full database (Supplement A), for each year an abridged version of the GeoPackage is also being shared that contains a subset of commonly used metrics for forest inventory (Supplement B)."***

**Both databases can be downloaded from Zenodo: *"The 2023 lidar plots and point cloud metrics described here are available at https://doi.org/10.5281/zenodo.16782860 on Zenodo (Bater et al., 2025)."***

(3) The accuracy of the ALS data collection (horizontal and vertical accuracies of points) and the classification accuracy of the point clouds are not provided or evaluated. This is an important factor to consider in order to know how reliable the datasets and the derived metrics are. Additional evidence needs to be provided.

**Because we are collecting data over remote landscapes which lack GNSS base station infrastructure, we did not require ALS vendors to differentially correct their data, nor did we require them to provide survey-based validation of bare-Earth terrain heights typical of ALS acquisitions in populated areas. Rather, ALS vendors employed precise point positioning (PPP) to post-process ALS return coordinates, with the expectation that they would obtain sub-metre horizontal and vertical accuracies. In the manuscript text, we write: *"The ALS vendors corrected GNSS data using PPP and all reported sub-metre horizontal and vertical accuracies."***

**Regarding the classification accuracies, we manually interpreted sampled point clouds to assess quality. The methods for this are outlined in section 6.4.5 of the Canadian airborne Lidar data acquisition guideline (*Natural Resources Canada and Public Safety Canada, 2022.*), but in short, the process involves clipping randomly selected 20 x 20 m areas and manually assessing the point cloud classifications. We include this information in the manuscript text: "*Point cloud classifications were validated by randomly selecting 20 x 20 m areas which were then clipped to perform three-dimensional checks.*"**

*Natural Resources Canada and Public Safety Canada, 2022. Federal airborne LiDAR data acquisition guideline Version 3.1 (No. General Information Product 117e), Federal Flood Mapping Guidelines Series. Government of Canada.* *https://doi.org/10.4095/330330*

More detailed comments are as follows:

1. L23. 15 million 900 m2 lidar plots, what is the point density?

**Good catch. The point density is a minimum of 12 pulses/m$^2$. As this line is from the abstract, we had to be parsimonious regarding information provided. Table 1 summarizes the acquisition specifications.**

**We will add this sentence to the abstract: *"Acquisition specifications included minimum swath widths of 500 m (year 2023) or 800 m (2024 and 2025), with a minimum pulse density of 12 pulses/m$^2$."***

2. L25. The authors mentioned that the chosen 30 by 30 m resolution was to match the medium resolution of Landsat and Sentinel 2, still, I believe lidar metrics with finer resolution should be considered, given the pulse/point density of ALS data, and it will be more beneficial for a broader user community. There are already country-wide lidar metrics derived from ALS data at 10 m resolution (see references below). 30 by 30 m resolution may be a bit limited in finer scale ecological applications. \

Ref: (1) Assmann, J. J., Moeslund, J. E., Treier, U. A., & Normand, S. (2022). EcoDes-DK15: high-resolution ecological descriptors of vegetation and terrain derived from Denmark's national airborne laser scanning data set. Earth System Science Data, 14(2), 823-844. doi:10.5194/essd-14-823-2022

(2) Shi, Y., Wang, J., & Kissling, W. D. (2025). Multi-temporal high-resolution data products

of ecosystem structure derived from country-wide airborne laser scanning surveys of the Netherlands. Earth Syst. Sci. Data, 17(7), 3641-3677. doi:https://doi.org/10.5194/essd-17-3641-2025

**We appreciate that rasterization and area-based analyses can be performed at a range of spatial resolutions and that 10 to 20 m is common for natural resources applications. The examples shared represent nations with a terrestrial area that corresponds to less than half a percent of Canada's land area. We used 30 m because it links directly to national satellite information products available in Canada. The ALS point cloud data will be made publicly available in the near future; users can then generate metrics at any scale they choose. We will add these elements to the discussion (and cite both papers). The suggested papers provide useful information, with your paper sharing useful details on what different metrics capture.**

3. L43-47. Too long for one sentence. Please rephrase.

**Thanks for noticing this. The sentence will be rephased as follows:**

***"It is not entirely fair to compare satellite remote sensing and ALS due to their differences in data costs to the end user, the level of detail captured, and the intensity and repeatability of collection. However, ALS provides access to simultaneous measurements of the vertical distribution of vegetation and the underlying terrain morphology (Lefsky et al., 2002), providing critical information on forest complexity and condition that cannot be obtained through other remote sensing methods."***

4. What is the impact of the ALS transects sampling discussed here on areas beyond Canada? It should also be reflected in the Introduction.

**We can expand on how ALS transect sampling can be used to inform estimates of vegetation attributes over large areas either through stand-alone sampling or in combination with other datasets (e.g. ground plots, wall-to-wall imagery). As an idea for the final manuscript, we can also include examples of systematic global samples of lidar collected by IceSAT-2 and GEDI.**

5. L66. It is not clear what "linear samples" mean here.

**Thanks for catching this. We will rephrase to: *"....collected as sampled linear ALS transects to extend....."***

6. L107-108. Please briefly explain how the current study relate/compare to the existing work mentioned here.

**Thank you for this suggestion. The next two sections (*1.1 Motivation* and *1.2 Objectives*) link directly to the literature review preceding these sections.**

7. Table 1. What are the horizontal and vertical accuracy of the acquisitions? What are the data volumes?

**Because we are collecting data over remote landscapes which lack GNSS base station infrastructure, we did not require ALS vendors to differentially correct their data, nor did we require them to provide survey-based validation of bare-Earth terrain heights typical of ALS acquisitions in populated areas. Rather, ALS vendors employed precise point positioning (PPP) to post-process ALS return coordinates, with the expectation that they would obtain sub-metre horizontal and vertical accuracies. In the manuscript text, we write: *"The ALS vendors corrected GNSS data using PPP and all reported sub-metre horizontal and vertical accuracies."***

**We address database volumes in the manuscript text: *"For the 2023 ALS transects, 15,353,866 lidar plots were generated within the lidar swaths. The full database including 369 point cloud metrics is 60.2 GB in size, and the abridged version of the database containing a subset of 40 metrics is 7.2 GB. Both versions are shared as SQLite GeoPackages."***

8. L185. Why remain low points noise (class 7) in the process?

**Class 7 (low points – noise) is a valid ASPRS las class and the returns contain useful information, especially for modeling vegetation regeneration. We do remove returns that are below 0 m relative to the DTM, and we will add that information to this section. Further, we offer users the option to select returns >0 m or > 2m depending on their needs.**

**We address height thresholds in the manuscript text: *"As the final products are intended to inform a variety of applications, including forest inventory, regeneration assessment, and wildfire fuels, the metrics were generated in four groups using: (1) all returns above 0 m, (2) first returns above 0 m, (3) all returns above 2 m, and (4) first returns above 2 m. Two height thresholds were used so that models could be created that either consider all vegetation from the ground surface upwards (i.e., ≥ 0 m), or with a focus on overstory structure (> 2 m)."***

9. L204. What is the accuracy of the points classification? For instance, what is the extent of misclassification between ground points and low vegetation points? It is essential for lidar metrics calculation and for deriving further metrics/parameters.

**Regarding the classification accuracies, we manually interpreted sampled point clouds to assess quality. The methods for this are outlined section 6.4.5 of the Canadian airborne Lidar data acquisition guideline (*Natural Resources Canada and Public Safety Canada, 2022.*), but in short, the process involves clipping randomly selected 20 x 20 m areas and manually assessing the point cloud classifications. We include this information in the manuscript text: "*Point cloud classifications were validated by randomly selecting 20 x 20 m areas which were then clipped to perform three-dimensional checks.*"**

**Natural Resources Canada and Public Safety Canada, 2022. Federal airborne LiDAR data acquisition guideline Version 3.1 (No. General Information Product 117e), Federal Flood Mapping Guidelines Series. Government of Canada. https://doi.org/10.4095/330330**

10. L221. Are there any data collected in 2025? As in Figure 1 and Table 2, only 2023 and 2024 were listed as acquisition years. This information should be clarified at the beginning.

**At the time of writing we were in the planning phases of acquiring 2025 data. The 2025 data have now been collected but will not be delivered until 2026. Because we have not yet received final delivery, details are not yet available. We will add as much detail as possible when we complete the manuscript revision.**

11. I think Table 5 can be better presented as a Figure. For instance, a land cover map for each ecozone, showing land cover classes and their areas. Lidar plot area can be overlaid with the land cover map with indications of sampling intensity.

**Thank you for this suggestion. While we agree with you that a figure would certainly be more visually impactful, the table is meant to provide raw numbers on sampling intensity broken down by land cover class, lidar plot area, and sampling intensity with the precision that a figure cannot provide.**

12. L255. How many of those areas have been selected for validation? And what are the validation results? Why is the area of validation (20 by 20 m) different from the original sample area (30 by 30 m)?

**Approximately twenty 20 x 20 m samples are collected per data delivery. The classification validation data are completely unrelated to the 30 x 30 m lidar plots described in the manuscript as they serve different purposes.**

13. L259. How do you rasterize the scan angles, and what is the use of the rasterized layer?

**We rasterize the data using LAStools and the surfaces are used for quality control only. We do retain them for future use.**

**We will modify this section to read: "*Point clouds were also rasterized based on return class (Table 1) and hillshades were generated from the DTMs. Raster surfaces were then visually inspected to ensure specifications were met (e.g., water was properly classified, DTMs were representative of the bare-Earth surface). Similarly, return counts and scan angles were rasterized to ensure transects fell within the specifications for point densities and swath widths (Table 1). All raster products were generated using LAStools (version 2.0.4) .*"**

14. L263. Is this including the volume of the raw point cloud? Why only 2023?

**This does not include the volume of the point cloud and includes only the geopackages. We are still processing the 2024 data and have not received delivery of the 2025 data.**

15. L269. This may link to the classification of ground points and low vegetation points. It would be helpful to know the quality of such classification.

**While we agree that the quality of ground returns is important, the differences seen here are driven by combinations of height thresholds (0 m or 2 m) and return type (all vs. first returns).**

16. Figure 4. Why are there large areas of red color in the water area in the Canopy Cover (the third panel)? Those areas are not shown in the P95 layer, meaning they probably are assigned to the NA value (i.e. no returns observed or masked out as water body)? Then why do those areas occur in the Canopy cover layer, probably having 0 value (which may should be NA)?

**First, there is a temporal mismatch between the image (tile 1) and the lidar data (tiles 2-4), so that some vegetation is present on the western point bar that appears in the height and complexity panels (e.g. riparian shrub). The cover panel includes returns over water because there were returns generated over water but no returns > 2m, resulting in a cover value of 0%. It is a useful illustration of representation of the data, but we may have to rethink the design of this figure to avoid confusion.**

17. Figure 5. The y-axis label of 1e6 for the number of lidar plots seems wrong. The purpose of showing this figure is not very well-explained in the caption and in the main text.

**We will remove the 1e6. The purpose of the figure is to demonstrate that different vegetation metrics can be summarized at the ecozone level, and by extension vegetation attributes (e.g. biomass) related to those metrics. We will expand on that in the main text.**

18. Figure 7. Please elaborate on the multidecadal NTEMS satellite information products. And what are the main things shown in this figure? Can information such as how many plots were repetitively disturbed by fire be included in the figure?

**For additional clarity we will expand upon what is shared in the Figure 7 caption. Examining how many plots were repeatedly disturbed is beyond the scope of this particular paper, but this is a great question we can explore in following publications.**

**We summarize NTEMS in the manuscript text: *"These methods build upon earlier work by the National Terrestrial Ecosystem Monitoring System (NTEMS), which was developed to monitor Canada's forested ecosystems on an annual basis using consistent, nationally available datasets (White et al., 2014; Wulder et al., 2024). The NTEMS relies primarily on medium spatial resolution satellite data (initially solely Landsat, now augmented with Sentinel 2) time series, integrated with ALS transects and ground plots, to generate national information products characterizing disturbance, land cover, and forest structure (Hermosilla et al., 2016). The first national lidar transect dataset was collected in 2010 to support NTEMS product development (Hopkinson et al., 2011; Wulder et al., 2012a), and subsequent work has shown that combining these data sources enables spatially comprehensive estimates of both forest structure and derived attributes (Matasci et al., 2018a, b; Zald et al., 2016)."***

19. L320. Can a use case be demonstrated here?

**We can bolster this sentence by citing papers previously mentioned in the manuscript.**

20. L331. Here I am missing a bit more concrete suggestions for using the 369 metrics derived. For instance, giving guidance on how to select the most relevant metrics/metric types for different ecological applications (with use cases and examples).

**We believe that suggesting how users select metrics for use as predictor variables for modelling is beyond the scope of this paper. To your point we could cite a few key papers on variable selection, but we would hesitate to delve much deeper into the subject. To reiterate, we do include two versions of the database, including an abridged version with 40 metrics commonly found in the literature that are useful for modelling vegetation-related attributes.**